# Bioactive Substances and Skin Health: An Integrative Review from a Pharmacy and Nutrition Perspective

**DOI:** 10.3390/ph18030373

**Published:** 2025-03-06

**Authors:** Rafael Jesús Giménez Martínez, Francisco Rivas García, Joan Carles March Cerdá, Ángela Hernández-Ruíz, Martha Irene González Castro, María-Isabel Valverde-Merino, Felipe José Huertas Camarasa, Fuensanta Lloris Meseguer, Margarita López-Viota Gallardo

**Affiliations:** 1Department of Nutrition and Bromatology, School of Pharmacy, University of Granada, 18071 Granada, Spain; rafaelg@ugr.es; 2Municipal Health and Consumer Unit, Guadix City Council, 18500 Guadix, Spain; saludyconsumo@guadix.es; 3School of Health Sciences, Valencia International University, 46002 Valencia, Spain; 4Andalusian School of Public Health, 18011 Granada, Spain; joancarles.march.easp@juntadeandalucia.es; 5Biosanitary Research Institute (ibs. GRANADA), 18012 Granada, Spain; 6Biomedical Research Network Centre (CiberESP), 28029 Madrid, Spain; 7Department of Physiotherapy, Nutrition and Sports Sciences, Faculty of Health Sciences, Universidad Europea de Valencia, Paseo de la Alameda 7, 46010 Valencia, Spain; angelahr.investigacion@gmail.com; 8Faculty of Health Sciences, Miguel de Cervantes European University, C. del Padre Julio Chevalier, 2, 47012 Valladolid, Spain; 9School of Nursing and Nutrition, Autonomous University of Chihuahua, Campus 2, Chihuahua 31125, Mexico; mgonzalezc@uach.mx; 10Pharmaceutical Care Research Group, School of Pharmacy, University of Granada, 18071 Granada, Spain; 11Advanced Chemistry Laboratory, University of Almeria, 04120 Almería, Spain; fjjjose23@gmail.com; 12Department of Educational Development and Vocational Training, Andalusian Government, 18016 Granada, Spain; fllormes405@g.educaand.es; 13Department of Pharmacy and Pharmaceutical Technology, School of Pharmacy, University of Granada, 18011 Granada, Spain; mlvg@ugr.es

**Keywords:** skin, bioactive substances, nutrition, pharmacy, cosmetics

## Abstract

The skin is one of the largest and most important organs of our body. There are numerous factors that are related to skin health, including lifestyle factors, nutrition, or skin care. Bioactive substances from plant and marine extracts play a key role in skin health. The aim of this research was to compile the main evidence on skin and bioactive substances. An integrative review was performed, reporting the main findings according to PRISMA (2020). Thirteen search equations were developed. After the applications of the equations and the process of screening and selection of articles, 95 references were compiled. The main results related to bioactive compounds were classified into food-derived components, nutraceuticals, symbiotics, active substances of marine origin, and substances from plant extracts). There are several factors that indicate that the use of bioactive compounds are interesting for skin health, highlighting some dietary nutrients, substances obtained from plant extracts and metabolites of marine origin that, showing anti-inflammatory and antimicrobial effects, are related to the improvement of some skin conditions or are active principles for cosmetics.

## 1. Introduction

The skin represents the largest surface of our body and is one of the most important organs. It is a complex structure that acts as a protection barrier against physical, chemical and biological factors, while also performing thermoregulation, and immune and mechanical protection functions, among others, which provide it with a crucial role in maintaining our health.

Anatomically, the skin is made up of two layers. The epidermis is the outermost layer and is in direct contact with the exterior. It includes several strata conditioned by the cellular growth of the germinative layer, and it is also basically made up of connective tissue, keratinocytes, melanocytes and Merkel and Langerhans cells. Of all the strata, the stratum corneum is the most superficial part and, according to its thickness, it can be differentiated into thick skin (palms of the hands and feet soles) and thin skin. Essentially, the epidermis has a mechanical function and a microbiological protection function that guards the deeper layers of the skin from injuries and infections [1].

Under the epidermis is the dermis, which contains proteins (collagen, elastin, and reticulin), nerves, blood and lymphatic vessels, and skin appendages (sweat glands and pilosebaceous units) [2].

The main functions of the dermis include the transport of nutrients, elimination of waste products, regulation of pressure and temperature, as well as mobilization of immune system cells and contribution to skin pigmentation [2]. The hypodermis deserves special mention as it forms the deepest layer of the skin, made up of subcutaneous fat, and is the location of the skin appendages, which provide mechanical support, synthesis and storage of substances useful for the skin [1,3]. The supply of nutrients through food and the components used in cosmetic products take on a major role in the development and maintenance of the skin, overall acting as a set of bioactive substances that may have therapeutic potential in the prevention and/or treatment of various pathologies. The diversity of the chemical structures that bioactive substances may have influences their bioavailability and biological properties [1,2].

Nutrition and skin health is fundamental, as nutrition plays an important role, since many of the components that form part of skin and mucosal structures are essential nutrients. The supply of nutrients through food and cosmetic components plays a major role in the development and maintenance of the skin. Pharmacy studies, formulates, and manufactures cosmetic products for daily skin care as well as products used in different therapies. In this sense, there is feedback between pharmacy and nutrition, since a lack of nutrients that the skin needs can be provided by food, either naturally or through nutritional supplements, and through products made by the pharmacy. In turn, the pharmacy can incorporate nutrients into its formulations of products for skin health and specific pathologies.

Nowadays, although some very interesting reviews have been published on bioactive substances and skin health and skin care, to our knowledge, a general review encompassing most of these bioactive substances and their effects on the skin has not been conducted and the scientific evidence has not been systematically collected.

There is an interrelation between the roles of pharmacy and nutrition through dermocosmetic products and nutrition, since these disciplines need and complement each other; it makes no sense to incorporate into our daily lives a wide variety of cosmetic products with multiple benefits for the skin if the skin’s nutritional needs are not covered. Therefore, skin health requires a source of biologically active compounds with cosmetic and dermatological relevance that act in a synergistic manner [2,3].

The aim of this article is to perform a review with a systematic methodology to analyze in a narrative approach of the main bioactive compounds of food, plant, and marine origin that could be useful for the skin, both to ensure its proper functioning and development and for their role in certain conditions such as dermatitis, psoriasis, acne, rosacea, hyperpigmentation, alopecia, and infections, among others.

## 2. Method

### 2.1. Type of Review: Design

An integrative literature review was planned, which is one of the most comprehensive methodological approaches regarding reviews and which enables exhaustive exploration of a given subject in order to recognize the current state of knowledge as well as to point out any gaps in knowledge. It is therefore a study of data collection carried out from secondary sources by means of bibliographical survey. To increase the rigor of the integrative review, the research follows six phases: drafting of the guiding questions, literature review, data collection, critical analysis of the included studies, discussion of the results, and presentation of the integrative review [4].

### 2.2. Data Sources and Search Strategy

The guiding questions were as follows: “Are there bioactive components of interest for skin health?”; “What are the main sources of such bioactive compounds?”. To answer these questions, an online bibliographic search was conducted in the following databases and covering the last five years: MEDLINE (PubMed), Scopus, Embase, and Cochrane Library.

The MeSH descriptors and subject-matter keywords were as follows: “Skin care”, “Skin”, “Nutrients”, “Nutrition”, “Pharmacy”, “Bioactive substance”, “Probiotics”, “Prebiotics”, “Cosmeceuticals”, “Diet”, “Marine compounds”. The same descriptors were used in English and Spanish.

Using these terms and adding the Boolean operators “AND” and “OR”, a complete and reproducible search strategy was established. The combination of different MeSH terms belonging to the same search equation required the “OR” operator, and “AND” was used to combine the different search equations. Certain filters were used to restrict the search, according to the purpose of this research, to studies in humans published in the last five years in either Spanish or English. Table 1 shows the search strategy used.

### 2.3. Eligibility Criteria and Data Synthesis

The inclusion criteria for selection of articles were as follows: (a) articles written in English or Spanish; (b) articles published in the last 5 years; (c) articles that considered the relationship between bioactive compounds and their role in skin health; (d) access to full texts from any database used; (e) review articles, clinical trials, and randomized controlled trials; (f) in vivo in vitro and human articles were included. Articles not related to the topic of the study and studies performed on animals were excluded.

### 2.4. Study Selection Processes

The selection of articles included reviewing the title and abstract to carry out screening, subsequent reading of other publications and finally reading the full text of the studies selected.

### 2.5. Data Extraction

Three authors (F-RG, MI-GC, and F-HC) independently identified studies and performed screening, study selection, and data extraction. To ensure inter-rater agreement, the percentage by which the number of times the three reviewers who performed the assessment reached agreement on the same question was added, and then this figure was divided by the total number of data items considered. To ensure reproducibility and minimize bias, disagreements were resolved by discussion with a fourth author (M-LV).

## 3. Results and Discussion

Following application of the different search equations on all the databases, 828 articles were initially retrieved using the selected filters. After reading the titles and abstracts, 793 articles were selected; 558 were subsequently eliminated, leaving 235 articles. Finally, after applying the aforementioned eligibility criteria, 95 articles were selected (Figure 1).

### 3.1. The Skin as an Active Biological Structure

The skin is not a static structure and is in continuous change due to its interaction with different factors that condition transformation over time [5]. In this sense, there are several factors that make it a structure with its own biological activity:Biological factors. These factors include [6]:
-Physical condition of the skin. It can be affected by acids, alkalis, wounds, and various diseases that cause a defective stratum corneum, leading to an increase in percutaneous absorption.-Skin age. As the skin ages (Table 2), a reduction in blood flow to the adipose tissue in the hypodermis is observed, which causes flaccidity. Also, there is a reduction in fibroblasts, glycosaminoglycans, oxygen consumption, and the genesis of intracellular energy, which translates into a decrease in protein synthesis and a lower capacity for cell mobility between the different layers. As the skin ages, collagen replacement decreases due to the reduction in collagenase activity. In addition to all this, degradation of fibronectin is responsible for skin distension [5].

Without a doubt, the epidermis is the layer where the age of the skin is more perceptible. Thus, with the passage of time, the hydrophilic/lipophilic fraction binomial of the hydrolipidic layer changes, together with a decrease in the associated pH and sebaceous secretions [2].

-Cutaneous metabolism. The skin metabolizes steroid hormones, carcinogens, and some drugs. This metabolism can vary and determine the efficacy of certain cosmetic compounds applied topically.-Skin regions. Variations in skin permeability are related to the thickness and nature of the stratum corneum and the density of skin appendages. Thus, the permeability of cosmetic ingredients will depend on the intrinsic resistance to permeation per unit of overall tissue thickness.

B.Physicochemical factors. These factors are of great importance since they guarantee the structure and functioning of the skin, and they condition the technological development of cosmetic products [7,8]:

-Skin hydration. Hydration of the stratum corneum is one of the most important factors related to the penetration of substances through the skin. Specifically, the process of water retention in the stratum corneum depends mainly on the presence of hygroscopic components inside the corneocytes and the intercellular lipids, as well as their ability to form a protective barrier against water loss from the skin.-pH. The stratum corneum is very resistant to pH changes; in fact, only ionized molecules pass easily through lipid membranes.-Temperature. The diffusion coefficient decreases as body temperature decreases; therefore, anything that increases body temperature will affect the permeability of ingredients in cosmetic formulations.-Size and shape of molecules. Smaller molecules will permeate better than larger ones.

Both the structure of the skin and the factors that affect it are of great importance among the characteristics to be considered in the formulation and action of a cosmetic product from the perspective of efficacy and safety. A cosmetic product requires that its formulation and components allow it to interact and pass through the stratum corneum; in fact, the degree of penetration and efficacy will be conditioned by the physicochemical properties and the size of the particles [3,4,5,6]. What is aimed for in a cosmetic product is that it passes through the epidermis without reaching the dermis, thus avoiding a systemic action.

The components of a cosmetic product (Table 3) pass through the stratum corneum by passive diffusion and can penetrate the deeper layers by various routes, among which the following should be noted [9]:(a)Transepidermal route, where molecules diffuse causing an accumulation of water that allows polar molecules to pass through the skin. However, an intercellular process can be used where the diffusion of the components is carried out through the lipid structures between the cells of the stratum corneum.(b)Transappendicular route, where the components of the cosmetic product use the skin appendages to migrate to the stratum corneum.

The best ingredients for the stratum corneum in a cosmetic product are considered to be those that are non-toxic, non-irritating, and hypoallergenic, with rapid and long-lasting action, without systemic pharmacological activity and with a reversible effect. In this sense, superficial penetration would be exerted by components such as vegetable and animal oils, butters, fatty esters, paraffins, vaseline, silicones, waxes, and cetyl and stearyl alcohol. However, ethanol and propylene glycol, glycerol esters, vitamins, urea, and amino acids would exert their action in the last portion of the stratum corneum without reaching the dermis [10].

This paper will briefly discuss the different groups of compounds that have shown a positive action on the skin and that can be found in foods that are part of our daily diet (vitamins, antioxidants, fatty acids), nutraceuticals, prebiotics, and algae. All of them have their application in pharmacy and nutrition.

### 3.2. Bioactive Compounds of Interest for Skin Health

Food-derived components

Diet includes a series of nutrients that, when administered through food or as dietary supplements, improve skin health and function:-Vitamin A

Vitamin A is a fat-soluble vitamin found mainly in liver, fish oils, butter, eggs, meat, fish, and foods of plant origin. Retinol has been shown to absorb UV radiation between 300–350 nm, which confers a protective role to the skin [6,10,11]. Regarding its role in the skin, it has been observed that: (a) it stimulates the synthesis of keratinocytes, proteins of interest for the epidermis, and collagen to prevent its degradation [11]; (b) it promotes blood circulation and angiogenesis in the dermis [11]; (c) it regulates exfoliation and achieves a 60% depigmenting effect [12]; (d) it protects against ultraviolet radiation [13]; (e) by reducing the production of sebum, it limits the growth of Propionibacterium acnes and protects against acne [14]; (f) it increases the thickness of the skin with an important mechanical protection role [13]; (g) it participates in the degradation of melanin, and therefore in skin color [15].

-Vitamin E

Vitamin E includes a group of compounds composed of α-, β-, γ-, and δ-tocopherol and α-, β-, γ-, and δ-tocotrienol. Vitamin E can be obtained through a diet rich in vegetable oils from seeds, spinach, nuts, and cereals. Both tocopherols and tocotrienols play an important role in maintaining skin health since they participate in the differentiation of keratinocytes [16], and they stimulate the synthesis of ceramides, collagen and elastin in order to maintain and preserve the lipid structure of the stratum corneum [17,18]. All this is combined with a protective role against erythema caused by ultraviolet radiation [19].

-Vitamin C

The absence of the L-gulonolactone oxidase enzyme forces humans to ingest vitamin C through food, mainly citrus fruits, vegetables, and greens. Among other functions, vitamin C participates in the differentiation of keratinocytes, collagen and ceramide synthesis, and acts as a photoprotector against aging [20,21,22]. A good combination of vitamin A and E has been observed in humans due to their characteristics against UV-A radiation [13,18].

-Selenium

Selenium is a mineral abundant in foods of plant origin (nuts, cereals, legumes, vegetables) and of animal origin (meat, fish, seafood, eggs, dairy products, and derivatives) that has a protective effect on the skin against oxidative stress caused by ultraviolet radiation. Its role is exerted by stimulating the activity of selenium-dependent antioxidant enzymes such as glutathione peroxidase and thioredoxin reductase [23]. Also, according to studies by Alehagen et al. [24], selenium can inhibit formation of wrinkles and act as a skin anti-aging agent due to its restorative action against damage caused by ultraviolet radiation.

-Zinc

Zinc intake through food is guaranteed by the intake of meat, fish, milk and dairy products, nuts, plant seeds, sunflower seeds, legumes, and cereals. The distribution of zinc in the skin varies according to the stratum considered and thus, in the epidermis, zinc is more abundant in the stratum spinosum, while in the dermis, the concentration of zinc is higher in the more superficial parts with respect to the deeper areas [25].

This mineral participates in the immune function of the skin by modulating the activity of macrophages, neutrophils, and promoting phagocytosis to inhibit inflammatory cytokines, an aspect that affords prevention of inflammatory skin lesions [26]. Skin formulations containing 10–25% zinc have shown increased effectiveness on dermatological conditions (keratosis, rosacea, and eczema) over a period of four weeks to four months [12,23,25].

Studies by Searle et al. [27], Glass et al. [28], and Kim et al. [29] have highlighted the role of zinc in numerous skin pathologies, such as acne vulgaris, rosacea, atopic dermatitis, and melasma, which are treated with cosmetic products containing zinc as an ingredient. In fact, zinc preparations in many cosmetic products eliminate excess sebum from the skin, restore its natural pH and have astringent, anti-inflammatory, and anti-acne effects; notwithstanding, due to its ability to reflect and disperse ultraviolet rays, zinc oxide is incorporated as a physical filter in sunscreens.

-Copper

Collagen maturation, melanin synthesis, and an antimicrobial action represent the main contributions of copper to the skin, as described in studies by Trompette et al. [30] and Pilkington et al. [2]. Therefore, apart from the copper contribution obtained through the intake of meat, fish, nuts, legumes, and cereals, copper can also be incorporated as an active component of facial creams due to its ability to penetrate the skin.

-Silicon

Silicon is a diet component found in cereals, vegetables, bananas, nuts, and legumes, and is present in the skin, hair, and nails; its proportion decreases with age. Among its contributions to the skin, silicon plays a role in the synthesis of collagen, elastin, and keratin, as well as in improving the structure of the skin and its mechanical properties (increasing the resistance as well as the thickness of nails and hair) [31]. Silicon is also incorporated as part of the silicones that are included in cosmetic products such as hair conditioners, shampoos, and facial creams to improve the strength and elasticity of the skin due to its smoothing characteristics [32]. In this sense, 600 mg/day improves skin texture and moisture [31].

-Polyunsaturated fatty acids

Polyunsaturated fatty acids of the ω-3 series (alpha-linolenic acid, docosahexaenoic acid, and eicosapentaenoic acid) and ω-6 series (linoleic acid, gamma-linolenic acid, and arachidonic acid) have great dietary and cosmetic importance. Studies by Trompette et al. [33], Thomsem et al. [34], Nolan et al. [35], Wilson et al. [36], and Michalak et al. [37] have shown the importance of these fatty acids in the lipid composition of the dermis; in maintaining the equilibrium necessary for an adequate water balance of the skin; in the permeation of active substances; in the provision of antiallergic properties; and in the promotion of cell regeneration, all together with their photoprotective role against ultraviolet radiation and their soothing effect against skin damage that causes irritation. It should not be overlooked that, due to their anti-inflammatory nature, dietary supplementation and the addition of polyunsaturated fatty acids as ω-3, cosmetic ingredients would provide a potential benefit for pathologies that present themselves with skin inflammation [38].

Food provides polyunsaturated fatty acids through vegetable oils obtained from fruits, seeds, and nuts. In turn, these vegetable sources can be used as a cosmetic ingredient for incorporation thereof; however, their suitability and role in the skin will depend on the percentage of saturated and polyunsaturated fatty acids. In this regard, borage, evening primrose, currant, and hemp oils are useful for products intended for dry, irritated, and atopic skin, while coconut, cumin, baobab, and neem oils will be more useful for oily and acne-prone skin. Michalak et al. [37] and Gade et al. [38] have shown that argan, sea buckthorn, green coffee, wheat germ, and fig oils are more useful for products intended for skin ripening, anti-aging, and photoprotection.

-Carotenoids

Carotenoids are compounds responsible for the color of a large number of plant foods, where they exhibit a greater variety and concentration, although they are also found in foods of animal origin, bacteria, algae and fungi.

Currently, more than 800 carotenoids are known, of which β-carotene, lycopene, lutein, zeaxanthin, astaxanthin, antheraxanthin, capsanthin, and β-cryptoxanthine are of note. Thus, vegetables and greens are a good source of lutein, β-carotene, violaxanthin and neoxanthin, while xanthophylls are usually found in greater proportion in fruits, although as described by Thompson et al. [39], carotenes, such as lycopene, can also be found in fruits such as tomatoes.

The scientific literature has collected the multiple functions of carotenoids in skin health, among which the following are to be noted: (a) modulation of lipogenase activity with anti-inflammatory action [39]; (b) conversion of retinoids to provitamin A [37]; (c) antioxidant and immunomodulatory activity [40]; (d) reduction of lipid peroxidation reactions in the skin [37]; (e) protection of collagen [25]; (f) suppression of damage caused by ultraviolet radiation (UVA A and UVA B) [41]; (g) reduction of skin erythema caused by action of ultraviolet radiation [42]; (h) protection against damaged skin [43]; (i) improved skin hydration, texture, and elasticity [44]; (j) reduced skin hyperpigmentation processes [45]; (k) supply of growth factors necessary for cellular regeneration of the epidermis and dermis [44]. Studies have shown that a carotenoid supplementation at a concentration of 2–3 mg/day has a protective effect against UV radiation [40,41].

-Polyphenols

Polyphenols are a variety of compounds that provide color to many plants and fruits. The most abundant food sources of polyphenols are spices (clove, star anise, mint, oregano, sage, thyme, spearmint, and rosemary), fruits (berries, plums, cherries, strawberries, raspberries, grapes, apples, peaches, apricots, nectarines, and pears), plant seeds (flax and soybeans), nuts (chestnuts, walnuts, hazelnuts, and almonds) and vegetables (artichoke, chicory, onions, spinach, broccoli, asparagus, and lettuce). Based on their chemical structure, polyphenols are broadly classified into stilbenes, phenolic acids, flavonoids, and tannins [46].

Polyphenols are used in cosmetic products for their moisturizing, softening, soothing, and astringent action, as they inhibit the activity of collagenase, elastase, and hyaluronidase [47]. In addition, they soothe irritation and reduce skin redness, accelerate the natural regeneration of the epidermis, and improve blood microcirculation and skin elasticity. Finally, they protect against ultraviolet radiation and are useful in skin pathologies such as alopecia, acne vulgaris, and hyperpigmentation [48,49].

Polyphenols are characterized by their antioxidant role against the numerous reactive oxygen species that the skin is exposed to and that cause premature aging, lipid oxidation, loss of antioxidant capacity, genesis of inflammatory reactions, collagen breakdown, and skin diseases [50]. Fernandes et al. [51], Song et al. [52], and Michalak et al. [53] have identified different specific groups of polyphenols that benefit functioning of the skin: (a) flavonoids that act by protecting against ultraviolet radiation, used in anti-aging, anti-cellulite products for rosacea and as depigmenting agents; (b) phenolic acids and tannins with anti-cancer, anti-allergic, anti-inflammatory, anti-aging, antimicrobial properties, used as photoprotectors to prevent skin erythema; (c) stilbenes with a marked anti-inflammatory and anti-hyperpigmenting action.

B.Nutraceuticals

Body aging is a process determined by numerous stages and by the chronological process itself, which becomes more visible in the skin since it is the organ most exposed to external environmental conditions. The most visible signs of skin aging include, among others, wrinkles, lack of firmness in the skin tissues, changes in blood circulation, and hyperpigmentation [1,2,3].

The manifestations of aging affect all layers of the skin (epidermis, dermis, and hypodermis) and their origin can be endogenous (cellular metabolism) or exogenous (solar radiation, environmental pollutants, and climatic factors). However, the body has numerous defense mechanisms against the action of free radicals, which act as responsible for oxidative stress and the aging that the skin suffers (4).

There are currently various cosmetic products of interest for skin care and skin aging that are usually administered topically. Given the potential problems that some treatments may pose with respect to their action on the dermis and hypodermis, the oral route is being considered through use of skin nutrients that can be found in food supplements, and which are called nutraceuticals.

Since the origin of the term, nutraceuticals have been considered as components that are part of food or that can be incorporated into our diet as supplements and that provide benefits for the prevention and/or treatment of pathologies. In this sense, there are more and more studies that connect their beneficial actions and skin health [54]. Thus, among the nutraceuticals that enjoy scientific evidence, the following are to be noted:-Ceramides. Studies by Shamlou et al. [55] and Ohnuki et al. [56] show that ceramides play a fundamental role in maintaining the skin barrier due to their moisturizing action on the stratum corneum and viscoelasticity of the skin.-Plant extracts. Plant extracts are a good source of bioactive compounds (Table 4) of interest in numerous skin conditions such as dryness, eczema, acne, hyperpigmentation, damaged skin, and aging [57].

Also, *ginkgo biloba* and sage extracts, useful for hair growth, hair loss, and shine, as well as for the treatment of dandruff [56], have been observed to play a role.

-*Glycosaminoglycans.* Chemically speaking, these are non-branched polymers formed by repetitions of disaccharides, composed of an amino sugar and uronic acid. They have a moisturizing function, since they increase the levels of collagen and hyaluronic acid while reducing the activity of metalloproteinase, with the consequent inhibition of collagenase and elastase [57].-*Collagen supplements.* Useful for fighting sagging skin and wrinkles due to their wealth of amino acids such as proline, glycine, and hydroxyproline [58].-*Coenzyme Q10*. This acts as a lipophilic compound that eliminates free radicals and intervenes in activation of the inflammatory process, with an antioxidant action. Because skin aging leads to a reduction in the concentration of coenzyme Q10, its use has an important role in restoring skin functions [37].

Based on the above, it could be said that nutraceuticals are useful components in cosmetic products given their moisturizing, antioxidant, photoprotective, and anti-aging role, as well as for maintaining the integrity of hair, nails, and skin elasticity. The main advantage of nutraceuticals would be their ability to reach areas of the skin that are difficult to reach topically.

C.Symbiotics

Symbiotics are defined as compounds that comprise a combination of prebiotics (substances that encourage the growth of the intestinal microbiota after being fermented by intestinal bacteria) and probiotics (compounds that contain defined and viable microorganisms to exert beneficial effects on the intestinal microbiome) [59].

The role of symbiotics cannot be understood without first considering the importance of the skin’s natural microbiota, whose function is to reinforce immune activity by producing antimicrobial peptides that modulate the inflammatory response. Therefore, colonization of the skin by pathogenic bacteria must be prevented. In fact, dysbiosis of skin microbiota is related to the genesis of acne, psoriasis, and atopic dermatitis [59,60]. Independently of skin microbiota, there are other factors (use of cosmetic products, tobacco, antibiotics, temperature, ultraviolet radiation, or humidity) that can generate cutaneous dysbiosis [59].

In recent years, the role that probiotics and prebiotics administered orally may have in the treatment and/or improvement of various skin conditions has been investigated (Table 5). In this sense, and in light of the numerous studies that have shown their beneficial effects on skin health after their oral and/or topical administration, the disciplines of cosmetics, nutrition, and pharmacy have begun a search for and joint integration of symbiotics as a useful resource for numerous skin conditions, although the mechanism of action varies depending on the route of administration. Notwithstanding, although research on skin microbiota has evolved in recent years, more robust clinical trials are still required to consolidate the possible protocol for use of active components within the field of dermatology.

D.Active substances of marine origin

Algae

Algae, classified as macro or microalgae, encompass a group of species that have shown diverse applications in the field of nutrition and pharmacy due to their health and skin care benefits. The interest in algae is based on the wealth of substances they produce under the temperature, salinity, osmotic pressure, photooxidation, and ultraviolet radiation environmental conditions in which they live.

The scientific literature lists numerous functions that different substances from algae provide to the skin:-Immunomodulatory function. *Microalgae* and *cyanobacteria* (*Rythrospira*/*Spirulina* spp., *Chlorella vulgaris*, *Chlorella pyrenoidosa*, *I sochrysis*, *Pleurochrysis cartae*, *Dunaliella*, *Porphyridium purpureum*, and *Rhodosorus marinus*) have amino acids with immunomodulatory activity. In cosmetics, algae ingredients are used for the treatment of a wide range of skin diseases. The immunomodulatory mechanisms whereby they exert their action would be on cytokines, interferons, interleukins, and tumor necrosis factors, which are secreted mainly by macrophages, lymphocytes, and keratinocytes in the epidermis [64].-*Antioxidant function.* Antioxidants obtained from microalgae are chlorophyll, vitamins, flavonoids, polyphenols, sterols, carotenoids, and vitamins A, B_1_, B_2_, B_6_, B_12_, C, and E that can be used as moisturizers and sunscreens, and to prevent and treat multiple skin conditions [65]. It has been observed that numerous components produced by algae such as chlorophyll, isoprenoids, tocopherols (ascorbic acid, ubiquinol, hydroxyanisole, fucosterol, and fucoxanthin), and algae carbohydrates prevent the formation of superoxide anion and hydrogen peroxide radicals that attack cell membranes and genetic material [66,67,68].-*Photoprotective function.* Microalgae have numerous mechanisms for repairing of genetic material as a result of photo/darkness, antioxidant systems, and protection from ultraviolet radiation that have been developed in their natural habitat [69].-*Skin regeneration and hydration.* Bioactive compounds obtained from microalgae are being incorporated into the cosmetic industry for their ability to stimulate collagen synthesis and as substitutes for hyaluronic acid. In this sense, hyaluronic acid provides an improvement of the extracellular matrix, internal skin homeostasis, skin hydration, and wound healing by promoting cell migration. It is highlighted that Kojic acid can be obtained from different types of mushrooms and is a by-product of fermented soy sauce and rice wine. It plays an important role as an antimicrobial, and in preventing sun damage, depigmenting, and age spots [69,70,71].-*Wound healing activity and anti-inflammatory action.* Numerous fatty acids from different marine algae, such as eicosapentaenoic acid, inhibit IL-8 and TNF-α [72], which has been shown in studies by Alipoor et al. [73] with respect to docosahexaenoic acid. Studies by Usoltseva et al. [74] and Luthuli et al. [75] on fucoidans, of a polysaccharide nature, have also shown a reduction in neutrophil adhesion and inhibition of proinflammatory proteins.-*Anti-aging action.* Zouboulis et al. [76] have shown how some species of cyanobacteria prevent photodamage, an aspect in line with that shown by the photoprotective action exerted by certain components of food and plant origin. *Sophorolipids*, used in cosmetic formulations to reduce skin thinning, also show an anti-aging action [77].-Anti-acne. *Sargarfuran* spp. algae have shown an inhibitory action on *Propiniobacterium acnes* [78].

Table 6 shows some bioactive substances that are used as moisturizers and promote a supposedly regenerative activity of the skin. The role of substances obtained from algae as excipients and vehicles for molecules or drugs should not be overlooked either, which increases expectations about their great contribution in the search for more natural alternatives in cosmetic products:-*Exopolysaccharides.* Used for their role as emulsion and foam stabilizers and their moisturizing capacity [79].-*Agar and carrageenan*, obtained from *Gelidium cartilagineum*, *Gracilaria confervoides*, *Chondrus crispus*, and *Gigartina mamillosa*, are used for the manufacture of creams, body lotions, soaps, shampoos, hair conditioners, toothpastes, deodorants, shaving creams, perfumes, and makeup [80].

Lately, research is being conducted into the development of nanoalgosomes as new nanomaterials that could be used as novel natural delivery systems for high biological-value bioactive substances from algae (antioxidants, pigments, lipids, and complex carbohydrates), as well as bioactive biological molecules [81].

Bacteria

Numerous polysaccharides from marine bacteria produced by *Alteromonas macleodii*, *Pseudoalteromonas* sp., *Vibrio diabolicus* show anti-aging activity [82]. Other species such as *Paracoccus* sp., *Agrobacterium* sp. and *Flavobacteriaceae* sp. produce carotenoids (saproxanthin and mixol) that have a protective function against oxidation [83,84,85]. Studies by Ding et al. [86] and Wijaya et al. [87] showed that analogues of N-acylhydrotyrosine and thalassothalic acids isolated from marine bacteria *Thalassotalea* sp. showed antityrosinase activity which is useful for facial hyperpigmentation problems.

Fungi

*Phaeotheca triangularis*, *Trimmatostroma salinum*, *Hortaea werneckii*, *Aureobasidium pullulans* and *Cryptococcus liquefaciens* are marine fungi that produce secondary metabolites with anti-hyperpigmentation action of interest for the skin [88]. Other species such as *Thraustochytriidae* sp. produce squalene, which has emollient, anti-inflammatory and restorative properties [89], as well as various carotenoids (astaxanthin, zeaxanthin, and canthoxanthin) with antioxidant action. Of interest are the *Exophiala* sp. and *Acremonium* sp. species for their function against ultraviolet radiation and acne, respectively [90].

Sea sponges

In vitro studies carried out with marine sponge of the *Halidona*, *Phorbas*, *Phakellia*, *Stelliden*, *Acanthella cavernosa*, *Chandrosia ramiformis* species have shown antioxidant, anti-inflammatory, photoprotective, and anti-acne action [91].

Marine corals

Coral powder is used in topical application to provide minerals to the skin and to protect it from UV radiation. It is also used as an antioxidant, anti-aging, anti-acne, skin softener, as well as for the preparation of lipsticks and deodorants. Alves et al. [90] and Geahchan et al. [92] determined that only a few secondary metabolites of coral (diterpene glycosides and pseudopterosins) have shown anti-inflammatory, analgesic, antibacterial, anti-acne, and wound healing action [93].

E.Substances from plant extracts

Current cosmetics require components from plant extracts that have properties capable of influencing the biological functioning of the skin (Table 7) based on obtainable substances, allowing for progression of collagen and eliminating the volatile effects of free radicals, thus allowing optimal skin functionality to be maintained [94]. Essential oils in cosmetic products range from 0.1 to 1% per 100g to have an action on the skin [91,93].

### 3.3. Future Lines of Research

This integrative review may be a possible further advance in the comprehension of the relationship between the effects of bioactive compounds in relation to skin health, because the main scientific evidence has been compiled in a slightly systematic fashion in comparison with previous reviews conducted to date with the same global scope [37,48]. Nevertheless, the chosen design, an integrative review, answered the needs of this research [96]. This paper could be a potential future review of a broader scope more specifically focused on certain subgroups of bioactive compounds or perhaps even other reviews such as systematic reviews, which might investigate specific compounds and their effect on skin alterations.

Future research should focus on the importance of using these bioactive compounds in combination to enhance the synergistic effects of bioactive compounds with different mechanisms of action and have more beneficial effects on skin health. Additionally, it would be necessary to carry out research in which the effect of bioactive compounds can be studied from different perspectives, diet, supplementation, or facial care from cosmetics.

## 4. Conclusions

The skin indicates possible nutrient deficiencies that can be provided through the diet on a daily diet. In fact, many bioactive substances can be used as active ingredients, excipients, and additives to generate high-value compounds such as cosmetic products. Therefore, vitamins A and E, carotenoids, polyunsaturated fatty acids, zinc, copper, selenium, silicon, and polyphenols that have shown anti-inflammatory and antimicrobial properties that, together with protection and restoration of the epidermal barrier, ensure an adequate level of skin hydration and protection against ultraviolet radiation and skin cancer. Ceramides, glycosaminoglycans, and carrageenans improve dry skin, eczema, acne, and damaged skin, as well as are useful constituents for skin and hair care. On the other hand, metabolites of marine origin algae, such as polysaccharides, mycosporine 2-glycine, fatty acids, and collagen have shown important anti-aging, anti-inflammatory, and anti-acne activity. However, plant extracts are used for their fatty acids and their role in skin care rather than disease prevention, hence their use in pharmaceutical products. The important role played by symbiotics against atopic dermatitis, acne, psoriasis, and seborrheic dermatitis should not be overlooked, although a greater number of studies is still required to reinforce the existing evidence.

## Figures and Tables

**Figure 1 pharmaceuticals-18-00373-f001:**
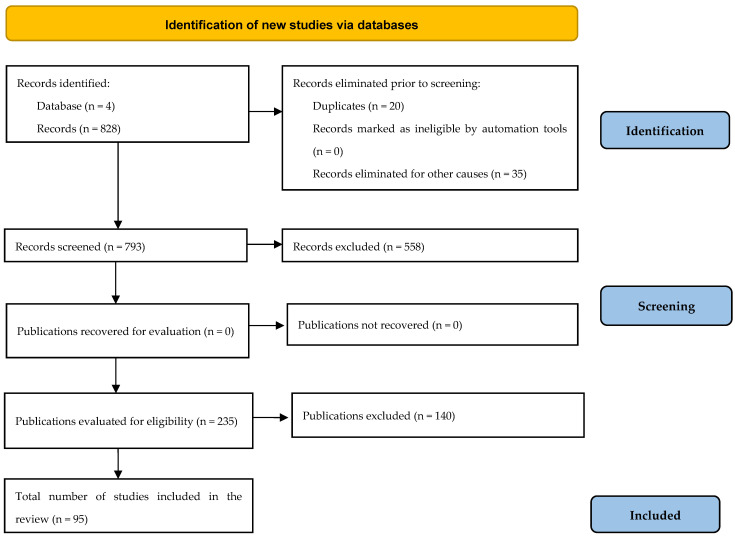
Flowchart for selection of articles according to the PRISMA Statement (2020) https://www.prisma-statement.org/prisma-2020-flow-diagram, accessed on 15 December 2024.

**Table 1 pharmaceuticals-18-00373-t001:** Search equations for the different databases.

(“Nutrients” [Title/Abstract] OR “Skin” [MeSH Terms])
(“Nutrients” [Title/Abstract] AND “Skin” [MeSH Terms])
(“Skin” [Title/Abstract] AND “Nutrition” [MeSH Terms])
(“Skin care” [Title/Abstract] AND “Pharmacy” [MeSH Terms])
(“Skin care” [Title/Abstract] AND “Nutrition” [MeSH Terms])
(“Skin” [Title/Abstract] OR “Nutrition” [MeSH Terms]) AND “Cosmeceuticals” [MeSH Terms])
(“Skin care” [Title/Abstract] AND “Pharmacy” [MeSH Terms]) AND “Nutrients” [MeSH Terms])
(“Skin care” [Title/Abstract] AND “Prebiotics” [MeSH Terms]) AND “Nutrients” [MeSH Terms])
(“Skin health” [Title/Abstract] AND “cosmeceuticals” [MeSH Terms]) OR “Nutrients” [MeSH Terms])
(“Diet” [Title/Abstract] OR “Skin” [MeSH Terms])
(“Skin health” [Title/Abstract] AND “Marine compounds” [MeSH Terms])
(“Nutrition” [Title/Abstract] AND “Skin health” [MeSH Terms])
(“Diet” [Title/Abstract] AND “Bioactive substances” [MeSH Terms]) AND “Skin health” [MeSH Terms])

**Table 2 pharmaceuticals-18-00373-t002:** Skin structure and age-related changes.

Structure	Modifications
Epidermis	Reduction of ceramides, vitamin D, sebaceous secretions, as well as cell differentiation and regeneration [1,3] Decreased mechanoreceptor and baroreceptor capacity [2]
Epidermis-hypodermis interface	Fibronectin degradation [2]
Dermis	Decreased thermoreceptors and skin turgor [1,2]

**Table 3 pharmaceuticals-18-00373-t003:** Diffusion of cosmetic components in different layers of the stratum corneum.

Stratum Corneum Layers	Components
Upper portion	Vegetable and animal oils, butters, fatty esters, paraffins, petrolatum, silicones, waxes, and alcohols (cetyl and stearyl)
Lower portion (before dermis)	Ethanol, propylene glycol, glycerol esters, vitamins, urea, and amino acids

**Table 4 pharmaceuticals-18-00373-t004:** Bioactive components of plant extracts of interest for the skin.

Skin Condition	Vegetable Extracts
Dryness	Castor oil, Mango, Coconut oil, Sunflower oil, Olive oil, Aloe Vera, Oats [56,57]
Eczema	Turmeric [50]
Acne, pigmentation, and pimples	Artemisia, Basil, Pea, Pumpkin, Onion [57]
Distressed skins	Red Clover, Chamomile, Jojoba Oil [53]
Aging	Ginseng [57]

**Table 5 pharmaceuticals-18-00373-t005:** Role of symbiotics in prevention and/or treatment of skin conditions.

Skin Conditions	Symbiotic Action
Atopic dermatitis	*Bifidobacterium* spp., *Propionibacterium* spp., *Coprococcus* spp., *Blautia* spp., *Bifidobacterium longum* and *Eubacterium* spp.: increase production of short-chain fatty acids (decreased in this condition) [61]
Acne	*Lactobacillus acidophilus*, *Lactobacillus delbrueckii* subspecies bulgaricus and *Bifidobacterium bifidum* L, rhamnosus, *Lactibacillus delbrueckii*: reduce inflammation, improve or prevent skin lesions [62]
Psoriasis	*Streptococcus aureus* and *Streptococcus pyogenes*: modulation of the inflammatory response by modification of the m-TOR pathway that improves the intestinal microbiome [63]
Seborrheic dermatitis	*Lactobacillus* spp.: useful for reducing erythema, scaling, and seborrhea of lesions on the scalp [60]
Skin cancer	*Lactobacillus jonhsonii*: photoprotective effect [61]

**Table 6 pharmaceuticals-18-00373-t006:** Bioactive compounds from algae of interest for the skin.

Bioactive Compounds from Algae	Functions on the Skin
Polysaccharides	Moisturizers [69] Metabolic regulation of water distribution [71]
Fatty acids	Collagen promotion [73]
Collagen	Stimulation of collagen synthesis [79] Collagenase inhibition [70]
Amino acids	Increased procollagen C [80]
Mycosporine 2-glycine	Collagenase inhibition [81]

**Table 7 pharmaceuticals-18-00373-t007:** Plant extracts used in cosmetic and care products.

Vegetables Extracts	Major Components	Functions
Coconut oil	Lauric, myristic, picric acid	Moisturizing, hair protector [94]
Almond oil	Gadoleic and margaric acid	Emollient [94]
Olive oil	Tocopherols and sterols	Emollient, hair conditioner [95]
Sesame oil	Myristic acid	Antioxidant [95]
Castor oil	Ricinoleic acid	Facial cleanser, moisturizer, hair strengthening [94]
Beeswax	Ricinoleyl alcohol, myricin	Antioxidant, antibacterial, anti-inflammatory [95]
Carnauba wax	Fatty acid esters	Moisturizing [94]

## Data Availability

Data sharing not applicable.

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
