# Peer review of "Bioactive Substances and Skin Health: An Integrative Review from a Pharmacy and Nutrition Perspective"

_pharmaceuticals, 2025, doi:10.3390/ph18030373_

Round 1
Reviewer 1 Report
Comments and Suggestions for Authors
1. Title: Bioactive Substances and Skin Health: An Integrative Review from a Pharmacy and Nutrition Perspective
2. The manuscript provides a comprehensive review of bioactive substances and their effects on skin health. The integration of pharmacy and nutrition perspectives is a valuable approach, but the manuscript could benefit from clearer articulation of the connection between these disciplines.
3. The abstract effectively summarizes the content but could include more quantitative details from the key findings. Abstract needs to be revised, including the importance of the topic , about your review and the conclusion.
4. The PRISMA flow diagram is a useful visual.
5. While the manuscript covers an extensive range of topics, the flow of information can be improved. Clearer transitions between sections and a more structured narrative would enhance readability.
6. Currently, the manuscript contains a large volume of information that is not well-organized, making it difficult for readers to follow and understand. Reorganizing the content into logical sections with clear headings would improve accessibility.
7. The conclusion effectively summarizes the findings but does not sufficiently highlight future research directions.
Author Response
1.The manuscript provides a comprehensive review of bioactive substances and their effects
on skin health. The integration of pharmacy and nutrition perspectives is a valuable approach,
but the manuscript could benefit from clearer articulation of the connection between these
disciplines.
R:A paragraph has been added on page 2 establishing the possible general integration of the two disciplines
2. The abstract effectively summarizes the content but could include more quantitative
details from the key findings. Abstract needs to be revised, including the importance of the
topic , about your review and the conclusion.
R: The abstract has been modified
3. The PRISMA flow diagram is a useful visual.
R: The PRISMA diagram is included in the article
4. While the manuscript covers an extensive range of topics, the flow of information can be
improved. Clearer transitions between sections and a more structured narrative would
enhance readability.
R: The commentary is appreciated, although due to the structure of the text it is complex to
reformulate everything, although an attempt has been made to establish some small details
of style and organization that may help to better distribute the headings.
5. Currently, the manuscript contains a large volume of information that is not well-
organized, making it difficult for readers to follow and understand. Reorganizing the content
into logical sections with clear headings would improve accessibility.
R: Small changes marked in red have been introduced that may improve all the information.Also, it should be noted that it is difficult to go into more depth as the idea of this article is to contextualize the importance of certain compounds of interest for the skin, hence there is perhaps a feeling of not establishing a relationship between headings because sections are addressed that although they are part of a global are referred to different groups.
7. The conclusion effectively summarizes the findings but does not sufficiently highlight
future research directions.
R: The conclusions have been modified, which also address future perspectives.

Reviewer 2 Report
Comments and Suggestions for Authors
Dear authors, it is a good study but improves the abstract and conclusion.
Address all the comments an follow the same.

Author Response
The suggestions made on the article and detailed below have been incorporated:
-The table cannot be deleted because it contains the search equations and we
consider their inclusion a priority. Perhaps because they are taken directly from
PUBMED, this is the reason for the high level of similarity.
- The abstract has been modified
- Changed symbiotic to synergistic manner
- Changed developed by planned
- A new paragraph f) is added in the section of Eligibility criteria and data synthesis
- Table 2 has been deleted
- A little more detail about hyaluronic acid and kojic acid.

Reviewer 3 Report
Comments and Suggestions for Authors
The authors propose in the presented manuscript a systematization of some scientific information regarding the influence of some bioactive substances on the health of the skin.
The authors took into account a very large number of scientific articles in the field, describing in detail how they selected them.
In my opinion, however, the organization and presentation of the material does not rise to the desired scientific level. The information is too general, being very little useful for further research. I consider that the subject is too vast, too general, the topic should be narrowed down and only a few directions should be addressed, with more concrete details, which would bring a useful contribution from a scientific point of view. For example: the role of vitamins or natural compounds in cosmetic preparations or in oral administration, with detailed explanations of how these compounds work, the necessary doses, the mode of administration etc., scientifically documented.
There are many incorrect and incomplete statements that require more careful documentation (for example: some data concerning the active principles from plants – carotenoids, polyphenols, plant extracts, table 5, table 8, etc…).
Some observations:
- Table 2 – it is not necessary, the information can be found in the references list
- The involvement of vitamins in the action at the skin level is correlated with a documented dose and way of administration, these aspects are important for evaluating the effectiveness and safety of administering a compound
- The carotenoids are natural compounds with some important characteristics which conditions their effectiveness, aspects that should be described in such a scientific paper
- Polyphenols represents a structural class that includes many categories of active principles (phenolic acids, flavonoids/anthocyanines, coumarines, tannins, lignanes, stilbenes etc.), each with its particularities and his representatives compounds, who act differently. It is not correct to generalize the pharmacological activities for all types of polyphenols, only those important for the dermato-cosmetic field should be considered in this review, with examples of their concrete mode of action.
- Conclusions are irrelevant, and does not present concrete data about the subject addressed, which could be useful for a specialist who is doing a documentation or research in the field.
Author Response
Comments
The authors propose in the presented manuscript a systematization of some scientific
information regarding the influence of some bioactive substances on the health of the
skin.
- The authors took into account a very large number of scientific articles in the field,
describing in detail how they selected them.
- In my opinion, however, the organization and presentation of the material does not
rise to the desired scientific level. The information is too general, being very little
useful for further research. I consider that the subject is too vast, too general, the
topic should be narrowed down and only a few directions should be addressed, with
more concrete details, which would bring a useful contribution from a scientific point
of view. For example: the role of vitamins or natural compounds in cosmetic
preparations or in oral administration, with detailed explanations of how these
compounds work, the necessary doses, the mode of administration etc., scientifically
documented.
- There are many incorrect and incomplete statements that require more careful
documentation (for example: some data concerning the active principles from plants –
carotenoids, polyphenols, plant extracts, table 5, table 8, etc…).
Some observations:
Table 2 – it is not necessary, the information can be found in the references list
- The involvement of vitamins in the action at the skin level is correlated with a
documented dose and way of administration, these aspects are important for
evaluating the effectiveness and safety of administering a compound
- The carotenoids are natural compounds with some important characteristics which
conditions their effectiveness, aspects that should be described in such a scientific paper
- Polyphenols represents a structural class that includes many categories of active
principles (phenolic acids, flavonoids/anthocyanines, coumarines, tannins, lignanes, stilbenes
etc.), each with its particularities and his representatives compounds, who act differently. It is
not correct to generalize the pharmacological activities for all types of polyphenols, only those
important for the dermato-cosmetic field should be considered in this review, with examples
of their concrete mode of action.
- Conclusions are irrelevant, and does not present concrete data about the subject
addressed, which could be useful for a specialist who is doing a documentation or research in
the field. - Response: The comments made in this regard are welcome, and the following considerations are made in response:
- Table 2 has been deleted
- The conclusions have been modified.
- Some changes of interest have been made
- The article could be more concrete and perhaps more specific by going into some aspects in more depth, but the philosophy of the article is not to go into the different components studied but to establish the general basis of the components and their role in skin health. We believe that the reviewer's very accurate comments are appropriate for a systematic review and/or meta-analysis type of article. In this sense, our group is considering several future monographic articles in which only each of the groups of substances are addressed independently.

Round 2
Reviewer 1 Report
Comments and Suggestions for Authors
Authors have made all the necessary changes.
It can be accepted now.
Author Response
The authors are grateful for their work, contributions and considerations with the article.
Reviewer 3 Report
Comments and Suggestions for Authors
The authors made some improvements to the manuscript, but I consider that these are not enough to raise the scientific level of the work. In my opinion, the valorization of the scientific material could be done at a better level, which would be really useful to the researchers in the field.
The work still remains too general, with a superficial treatment of some aspects related to plant products and active compounds from extracts.
Author Response
We, the authors of this article, respect and positively accept their considerations. In this regard, we would like to comment that:
1.- Modifications have been made to the article, including the reviewers' suggestions.
1.- Perhaps the article may not appear to be of a high scientific level, but it has been written by professionals in the fields of Pharmacy and Pharmaceutical Technology, Nutrition, Dietetics, Chemistry and Pharmaceutical Care whose main aim has been to show in a general way how bioactive compounds that can come from food and/or be used in the field of pharmacy can have an effect on health.
2.- This article aims to contextualise the known role and evidence of the actions of certain bioactive compounds, which can be a starting point to inspire possible research into the general actions highlighted.
3.- It is not intended to expand and go into much detail on each of the components as the aim is to show in a general way the benefits for skin health.
Round 3
Reviewer 3 Report
Comments and Suggestions for Authors
The authors made some improvements to the manuscript, taking into account only a part of the recommendations.
I do not dispute the professional quality of the authors, I respect the work done, only that, in my opinion, a scientific article published in such a journal must presents clear information, well documented and defined from a scientific point of view. In particular, the part related to the pharmaceutical approach of plant resources with applications in skin health should be improved.